# Implementing New Technologies to Improve Visual–Spatial Functions in Patients with Impaired Consciousness

**DOI:** 10.3390/ijerph19053081

**Published:** 2022-03-05

**Authors:** Katarzyna Kujawa, Alina Żurek, Agata Gorączko, Roman Olejniczak, Grzegorz Zurek

**Affiliations:** 1Department of Biostructure, Wroclaw University of Health and Sport Sciences, 51-612 Wroclaw, Poland; katarzyna.kujawa@awf.wroc.pl (K.K.); agagoraczko@gmail.com (A.G.); 2Neurorehabilitation Clinic in Wroclaw, 54-530 Wroclaw, Poland; roman.olejniczak@gmail.com; 3Institute of Psychology, University of Wroclaw, 50-137 Wrocław, Poland; alina.zurek@uwr.edu.pl

**Keywords:** visual–spatial functions, impaired consciousness, brain damage, eye tracking, oculomotor training

## Abstract

The quality of life of patients with severe brain damage is compromised by, e.g., impaired cognitive functions and ocular dysfunction. The paper contains research findings regarding participants of an oculomotor training course aimed at the therapy of visual–spatial functions. Five male patients with brain damage who did not communicate, verbally or motorically, participated in the study. Over a six-week period, the subjects solved tasks associated with recognising objects, size perception, colour perception, perception of object structures (letters), perception of object structures (objects), detecting differences between images and assembling image components into the complete image with the use of an eye tracker. The findings present evidence of oculomotor training effectiveness based on a longer duration of the work with the eye tracker and improved visual–spatial functions.

## 1. Introduction

Brain damage is a serious public health issue, which affects individuals in all demographic groups [1]. Damage may be caused by an injury (traumatic brain injury, TBI) or a cerebrovascular disorder that results in an ischemic stroke (blockage of the cerebrovascular circulation) or a haemorrhagic stroke (a rupture of a blood vessel inside the brain) [2]. Stroke is one of the most common forms of acquired brain injury (ABI) [3]. The consequences of stroke usually include movement disorders, such as hemiparesis or paralysis of one side of the body and central facial paresis. Moreover, language and speech disorders, e.g., aphasia and dysarthria, are common. The effects of brain damage may also manifest themselves in various forms, from mild consciousness impairment to an unrelenting comatose state or death [4]. Other forms include visual impairment and disorders of cognitive functions (CFs). When brain damage results from a traffic accident, ocular traumas, spinal cord injuries and other injuries, e.g., pelvic or limb fractures, are also commonly observed [5].

In addition, patients frequently demonstrate signs of depressed mood. Many patients manifest impaired neurobehavioural functions, mainly associated with diffused axonal injury [6]. Therefore, brain injuries dramatically affect patients’ quality of life due to the effects of the brain damage and they are considered depression risk factors [7]. An incident of brain tissue damage is an onset of a chronic disease process that involves the whole body [8,9,10].

Due to cognitive disorders, many individuals who experience severe TBI require long-term rehabilitation and neuropsychological support [5,11,12]. Cognitive deficits are primarily the effects of white matter degeneration, negative neurochemical changes or cortical spreading depression (CSD) [5,13]. CSD is a pathology manifested by the expanding of depolarisation wave within the grey matter which disrupts the ionic homeostasis of neurons and glial cells, leading to an increased energy demand. This results in a metabolic crisis and further neuronal death. CSD is considered to be the major source of negative effects of brain damage [13].

A particularly great amount of time in the therapy of patients with brain damage is dedicated to the training and elimination of cognitive deficits [14]. Proper cognitive functioning is essential for these patients, as it is the basis for proper performance of specific daily activities where continuous choices of adequate strategies are necessary to achieve the goal [15]. Cognitive functions are the spectra of mental abilities and complex processes associated with attention, memory, judgment and assessment, problem solving, and decision-making as well as language understanding and linguistic synthesis [16]. All these processes are feasible due to cooperation of neuronal networks [17]. The most important networks described to date include the cingulo-opercular network, fronto-parietal network, bilateral anterior prefrontal cortices, dorsal anterior cingulate cortex and thalamic connections [18]. Despite a significant technological progress and access to neuroimaging techniques, a precise understanding of the ways that tasks are performed and controlled is still subject to study [15].

Visual–spatial functions (VSFs) are the CF components, and they are performed through a few separate cognitive processes. In the available literature, the VSFs contain six various components, the closure flexibility, figural fluency, spatial orientation, visualisation, spatial scanning and the visual memory. It is worth mentioning that the visual–spatial processes have been empirically identified as separate abilities in the factor analysis [19].

Cognitive function treatment methods largely differ depending on the severity of brain injury and range from radical surgery to daily sessions of cognitive therapy [4]. An interesting example of elimination of the cognitive deficit effects, with persisting mild functional deficits, in a group of TBI patients, was reported by Vas et al. The scheduled cognitive training contributed to the improvement of cognitive control, associated executive functions and daily function, as well as to improved mental health and facilitated positive neuronal plasticity in the study subjects [20].

Patients with brain injury require complex forms of therapy [21]. One of them is oculomotor training with the use of eye trackers (ETs). ET applications in the oculomotor training are very wide, ranging from activities aimed at improving the health of patients who only communicate by means of their eye movements or patients with progressive retinitis pigmentosa to applications regarding healthy individuals in, e.g., professional sports, the aviation sector or the automotive industry. The available research findings show that various forms of oculomotor training effectively improved the signs and symptoms of many clinical disorders, including post-stroke cognitive, gait or functional impairment [22]. The great importance of oculomotor training is also declared for non-invasive rehabilitation of ocular alignment defects [23,24]. Proper ocular control is essential, as it is one of the proper cognitive processing components [22]. Therefore, it is important to consider the eye movement therapy while planning therapeutic interventions in individuals with damaged brain. Thus, improvement of the ocular performance may be supported by the use of new technologies such as the ETs. The use of ETs is justified also by the fact that the oculomotor function is usually maintained in patients with a brain injury even in cases of markedly damaged brain tissue, so the use of these devices is possible [25]. Moreover, the eye tracking technology allows to collect reliable quantitative data [26,27,28]. In addition, eye movements engage many brain regions, which stimulate neuroplastic mechanisms [29]. The concept of neuroplasticity is associated with processes of brain reorganisation and remodelling, ranging from molecular, cellular and synaptic alterations to more global changes in interneuronal connections being responses to cognitive requirements, environmental stimuli, behavioural experience or an injury [30,31]. The fundamental principle of neuroplasticity is plasticity of synapses, which are constantly removed or restored and the equilibrium of these opposite processes largely depends on neural activity [32]. Neural plasticity is the basis for orthoptic rehabilitation of ocular neuromuscular disorders [33].

In view of the above, a question arises whether the VSF improvement is observed after the oculomotor training in patients with brain damage. The beneficial effects of such training could help develop early therapeutic interventions aimed at improving the ocular motor function and, possibly, further improvement of the cognitive functions. It is necessary to expand the research in this field, as its long-term goal is also to improve the quality of life of patients with brain injury.

The objective of the research was to compare the durations of active performance of visual–spatial function tasks with the use of an eye tracker before and after the oculomotor training course.

## 2. Materials and Methods

### 2.1. Characteristics of the Study Group

Twenty patients, aged 26 to 67 were found eligible for the study. Due to the pandemic limitations and death of some participants, it was impossible to collect the whole set of results from all the subjects (see Figure 1). The analysis of data was based on the results collected from 5 male patients with brain damage of various aetiologies and with different levels of consciousness impairment. The patient characteristics are presented in Table 1. The data were collected over the period of 2019 and 2020 in the Centre of Palliative Care in Będkowo, Poland. Inclusion criteria were: consent of the patient’s guardian to participate in the study, person after completion of standard medical care, lack of verbal and motor communication with the environment, brain damage of different ethology, at least one functioning eye-ball. The subjects participated in a 6-week oculomotor training course (solving tasks using the eyes), which involved performance of VSF-related tasks. The consciousness level of each subject was determined by a physician based on the Coma Recovery Scale Revised (CRS-R) score before inclusion into the study. Caregivers of all the subjects signed the written informed consent form and gave their permission to publish the study data in accordance with the principles of the Declaration of Helsinki. The research was approved by the Senate Committee on Research Ethics at the Wroclaw University of Health and Sport Sciences, Poland (Consent no 29/2017).

### 2.2. Research Instrument

The patients performed VSF tasks with a C-Eye Pro eye tracker. The device uses infrared radiation to locate the patient’s eyeballs. The radiation does not disturb the user’s performance because it is invisible. Before task commencement, the monitor was placed in front of the patient’s eyes at a distance of 50 cm and calibrated in order to determine the exact fixation point. During the calibration procedure, the patient’s task was to observe a red blinking dot with a white border placed in the middle of the screen. The system communicated the calibration correctness by changing the screen display: a cursor (a small red dot) appeared and corresponded to the fixation point. A separate catalogue was created for each patient in the ET system where the test results were automatically recorded.

### 2.3. Visual–Spatial Function Testing

The study subjects performed 23 VFS-related tasks classified into the following categories: (A) recognising objects, (B) size perception, (C) colour perception, (D) perception of object structures—letters, (E) perception of object structures—objects, (F) finding differences between images and (G) assembling pieces of an image into the complete image.

The A category contained three tasks involving shapes of objects. The device read the instruction, ‘Find the same object’ in each task. An object example was displayed in the upper part of the screen. Three other images were displayed in the lower part of the screen. The patient’s task was to select the correct image, i.e., the one which was the same as the example displayed above it.

The B category contained three tasks where the patient searched for the shape of an object which was consistent with the size of the example displayed. The device read the instruction, ‘Find the object of the same size’ in each task. The patient saw the same object shown in three different sizes in the lower part of the screen.

In the C category, an object of irregular shape and a random colour was displayed. The patient’s task was to select an object of the same colour from the objects of various colours displayed in the lower row. This category contained three tasks with the instruction, ‘Find the object of the same colour’.

The D category was connected with recognising letters (3 tasks). An incomplete capital letter was displayed in the upper part of the screen. Complete capital letters were presented below. The instruction for the D category tasks was, ‘What letter is this’? The patient’s task was to select the correct letter from the letters displayed in the lower row.

The E category tasks (3 tasks) involved the choice of an appropriate set of images. Overlapping contours of two different objects were displayed in the upper part of the screen. The lower row presented two images of separate contours of two different objects. One of the sets did not match the example displayed above it. The instruction for this task was, ‘What objects do you recognise’?

In the F category, the device read the instruction, ‘Show the differences between the images’. Two sets of white and black objects were displayed on the screen. The set example was presented on the left side while the right-sided set did not contain three components. In each of three tasks, the patients searched for three differences (missing components).

The last category (G) contained five tasks where three halves of contours of various objects were displayed in the middle of the screen. In each task, a half of the object contour which matched only one of the examples displayed was connected with the cursor. The device displayed the instruction, ‘Assemble the correct image’ and the patient’s task was to assemble both of the matching object halves into the complete image. Examples of tasks for each category can be found in the (Appendix A.

In addition, the overall duration of the patient’s work with the eye tracker was measured. It did not include any breaks, such as the times when the patient was sleeping or looking beyond the monitor, so the duration of the patient’s active work with the device was determined.

### 2.4. Oculomotor Training

Oculomotor training was conducted once a week for one clock hour; training was performed by experienced physiotherapist in ET technology. The exercises included in the training were performed only with the eye movements. Before performing the tasks, a one-point calibration was performed on each participant to optimise the functioning of the device. Participants performed tasks from the following categories: A, B, C, D, E, F, and G (see Section 2.3), which were presented in the same order for each participant. Participants were allowed to complete the tasks in each category for as long as necessary (there was no time limit in this aspect); however, the total time taken to complete tasks in all categories was taken into account in the analysis. After completing the tasks in the first category, the training physiotherapist switched the patient’s task set to next category.

### 2.5. Statistical Analysis

A statistical analysis of the results was performed in the Biostructure Research Laboratory at the Wroclaw University of Health and Sport Sciences, Poland (with ISO 9001 certification). The medians of the duration of active work with the ET were determined and the results were compared using the Wilcoxon matched pairs test with the significance level at *p* < 0.05000. The rates of properly performed tests were then calculated for all patients in each task category. The Wilcoxon matched pairs test was also used for the assessment of oculomotor training effectiveness measured by the rates of the tasks performed.

## 3. Results

In the study, durations of the VSF tasks before and after the oculomotor training course were compared. We demonstrated a statistically significant difference between the active work durations before (Test 1; B1) and after (Test 2: B2) the six-week training course. All patients were able to work actively with the device for a longer time and to solve more tasks provided. The durations of active work with the ET for individual patients are compared in Table 2.

In addition, percent rates of the performed VSF tasks were presented. The patients did best in the A category tasks (recognising objects) before participation in the project but the findings after six weeks of the work with the ET showed the highest rate of correctly performed tasks within the C category (colour perception). The most considerable changes following the training were observed for the B category (size perception). No changes were observed for the F category (finding differences between images).

The most difficult tasks for the patients before the project were associated with assembling pieces of an image into the complete image (the G category). However, the eleven-fold increase of the number of correctly performed tasks within this category is worth mentioning. The results classified into the task categories for the study group are presented in Figure 2.

The findings of the Wilcoxon matched pairs test showed only a 5% median value for the results of the initial tests. The further results following the 6-week oculomotor training course demonstrated a statistically significant improvement: the median value was 53% for the tasks provided to the patients (Figure 3, Table 3).

## 4. Discussion

Even a mild injury of the primary visual cortex leads to severe and long-term impairment of the ways that neurons encode visual information [34]. The impact of a brain injury on the visual system has been described by Frankowski et al. (2021). They observed a dramatically decreased activity, impaired responses to visual stimuli and spatial perception deficits of the primary visual cortical neurons [34]. Impaired visual perception is observed in approximately 75% of the patients with brain damage, and it frequently results in serious limitations concerning independent functioning [14]. These patients demonstrate visual field. There is some evidence suggesting that intense training following this first, spontaneous improvement may additionally expand the visual field by a few degrees [35].

Our research findings suggest that the ocular motor skills may also improve. The patients were able to work actively with the eye tracker for a longer time, which is supported by a statistically significant longer duration of work with the device following the six-week oculomotor training course. Moreover, better focusing on selected objects was observed among the patients after the training of ocular muscles. In addition, improved smooth pursuit was demonstrated in each subject after the research. Reduced numbers of saccadic pursuit movements were also observed, which suggests improved eye movement control in the patients following the oculomotor training course. Similar findings were observed in the research conducted by Murray et al. [22].

Observation of the patients allowed to discover that the subjects with initial problems with extreme saccadic movements (quick looking at the other side of the screen) were able to look in the direction they previously found difficult to see after completion of the oculomotor training course. In certain patients, difficulties with symmetric ocular movements result from, e.g., visual–spatial deficits (or neglect) [14]. The neglect is generally defined as inability to perceive, to report and to orient to sensory events towards one side of a space, contralaterally to the side of the lesion, with or without a primary sensory deficit [36,37]. Eye movement deficits observed during the initial tests may also have been associated with weakened ocular muscles [38]. These muscles are stimulated by signals from the cerebral and cerebellar cortices as well as the basal ganglia while the brainstem translates these instructions into motor nerve signals [39]. There are slow and fast eye movements. Slow movements refer to pursuit movements, i.e., smooth tracking of a moving object to keep it on the fovea. Fast eye movements basically correspond to saccades which rapidly change the point of fixation. Saccades do not require a visual target and may be performed in the darkness. Properly analysed saccades are sources of important information about the neurological state of a patient [39,40].

In our study, we evaluated the effects of the VSF training. All subjects achieved better final results in each VSF category. Differences between the initial and final tests, expressed by the rates of correctly performed tasks, were the most considerable for the B (size perception) and G (assembling of pieces of an image into a complete image) categories. On the other hand, the patients performed the largest number of tasks within the C category (colour perception) during the final tests. These findings suggest that patients with brain damage are mostly able to recognise colours.

The human colour vision is possible due to the presence of photoreceptors called rods and cones. A human eye is equipped with three cone types: (1) short-wave-sensitive cones responsible for detection of the blue colour; (2) long-wave-sensitive cones aiming at detection of the red colour; and (3) middle-wave-sensitive cones responsible for detection of the green colour. A combination of the three vision types is called trichromatic vision and allows to distinguish 10 million colours [41].

The research shows that the ability of colour vision contributes to better recognition of shapes [42] as the use of information stored in the memory is more effective in the case of tasks involving variously coloured objects. Moreover, colours may support the process of finding appropriate information because it is a property of the mnemotechnic representation of scenes and objects in the long-term memory. The colour knowledge results in better classification and recognition of objects [42], which has been confirmed by Gegenfurtner and Rieger (2000) who discovered that the study subjects better-recognised presented colourful objects compared to grey-coloured ones. Their conclusions suggest that the colour information provides additional perceptive clues allowing for determination of the form and structure of the scene being observed by the subjects [43]. It seems, therefore, that our findings confirm the literature reports. When the colour vision was maintained, the patients were able to solve the tasks within this category at a higher level, both in the initial and final tests.

The most difficult tasks for the patients at the initial stage of the project were associated with the assembling of pieces of an image into the complete image (the G category). However, they could cope successfully with 44% of these tasks following the oculomotor training course. We assume that the difficulties with recognition of the whole object while looking at only a half of it may be associated with both visual aphasia and visual agnosia. The concept of aphasia is known in the context of impaired understanding and formulating of statements caused by the damage of the cortical centre; it may also refer to impaired writing abilities [44]. In the case of visual aphasia, patients find it difficult to name the objects they are looking at but they know their applications; they do not manifest these problems during the tactile or verbal presentation [45]. Agnosia refers to impaired object recognition and it is frequently observed in neurological diseases. Patients can see the objects they are looking at, but they do not recognise them [46]. In the case of our patients, we can suspect a type of agnosia called simultanagnosia, i.e., the ability to recognise only components of images and the inability to see their relationships. As a result, patients do not perceive images as a whole [47]. However, observations of the patients and the tasks performed during the consecutive weeks of the oculomotor training course as well as significantly improved rates of the correct answers following the course may suggest that the improvement may also result from enhanced eye movement control. The patients were better able to recognise the components (two halves) of images and better able to control their matching, which was very difficult initially due to decreased eye movement control.

Another interesting finding during the initial tests was that the highest rate of correct answers was in the A category (recognising shapes). Shape recognition is managed by the inferior temporal cortex. The absence of this area compromises the ability to recognise patterns and shapes because local neurons have large receptive fields which extend to both parts of the visual field, including the representations of the central fovea [48]. The high rate of correct answers (40%) in our study may be explained by the brain’s ability to easily recognise whole shapes. The human visual system may rapidly encode the shape information using the retinal neural circuits [49]. Moreover, most theories of the object recognition mechanisms present data suggesting that a shape is the basic feature of object recognition and its retrieval from the long-term memory [50]. This rapid and basic analysis of the stimulus was probably one of the factors which helped the patients achieve the highest scores during the initial tests.

Studies on the effects of the object size on its shape recognition by humans are very limited [51]. During the initial tests, our patients had serious problems with matching the proper object size with the image provided as an example (only 13% of the correct answers). However, the results following the six-week training course demonstrated a significant improvement (up to 67%) and they were almost comparable to those associated with the A category (shape recognition). This may have been related to the improvement of both size recognition and eye muscle control.

Our research shows that eye muscle improvement may be achieved by the enhancement of their coordination and strength while using eye movements for solving tasks connected with one cognitive function (the visual–spatial function in our study). It seems particularly important in terms of new therapeutic options for patients with brain damage who do not communicate verbally or motorically.

## 5. Conclusions

The research findings suggest that the VSF-related oculomotor training may help restore a selected cognitive function and improve ocular motor skills. During the final tests, longer durations of the active task performance and increased numbers of correctly solved tasks within the majority of categories were observed. Following the oculomotor training course, the subjects coped best with the VSF category associated with colour recognition and the most considerable result changes were observed for the abilities to recognise shapes and to assemble image pieces into the complete image. Further research, additionally including the time it takes for the patient to search for the answer and the distance travelled by eyes before reaching the target, will contribute to a better understanding of mechanisms related to the therapeutic process regarding patients with severe brain injuries who may only communicate by means of their eye movements. Thus, new technologies may improve both the quality of life of these patients and the quality of the health service.

## 6. Limitation

The authors are aware of the fact that the limitation of this project is the small size of the study group. Individuals with severe brain injuries constitute a specific group of patients where a high drop-out rate may be expected, which is associated with the fact that their willingness to participate in research studies is affected by many factors, e.g., the weather, the emotional state, feeling unwell, and somnolence as well as an increased disease predisposition or a markedly higher risk of death.

## Figures and Tables

**Figure 1 ijerph-19-03081-f001:**
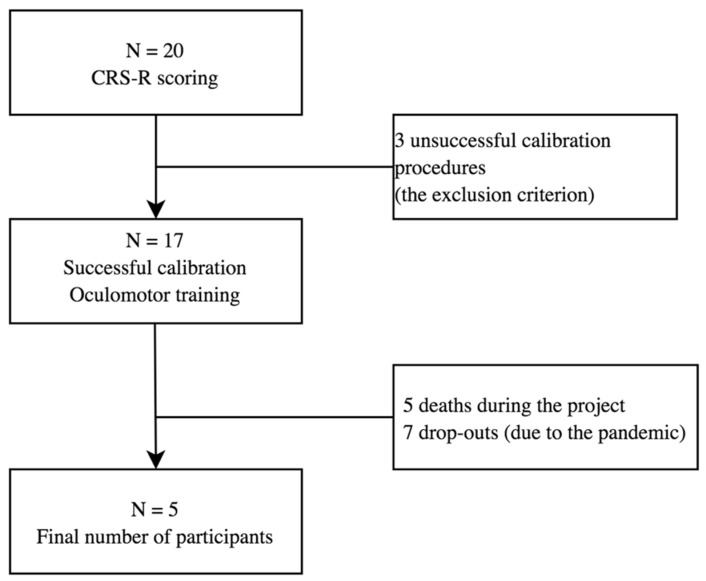
Flow chart. The process of study subject recruitment.

**Figure 2 ijerph-19-03081-f002:**
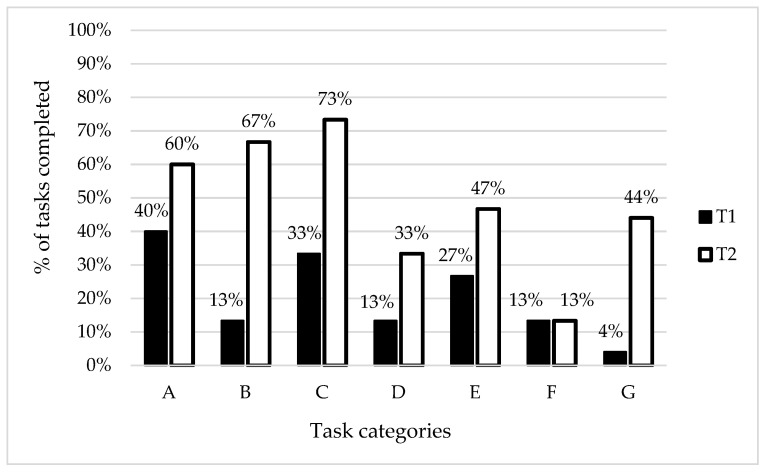
Study group findings classified into the task categories. Note: T1—results of the initial tests, T2—results of the final tests, A—recognising objects, B—size perception, C—colour perception, D—perception of object structures—letters, E—perception of object structures—objects, F—finding differences between images, G—assembling image pieces into the complete image.

**Figure 3 ijerph-19-03081-f003:**
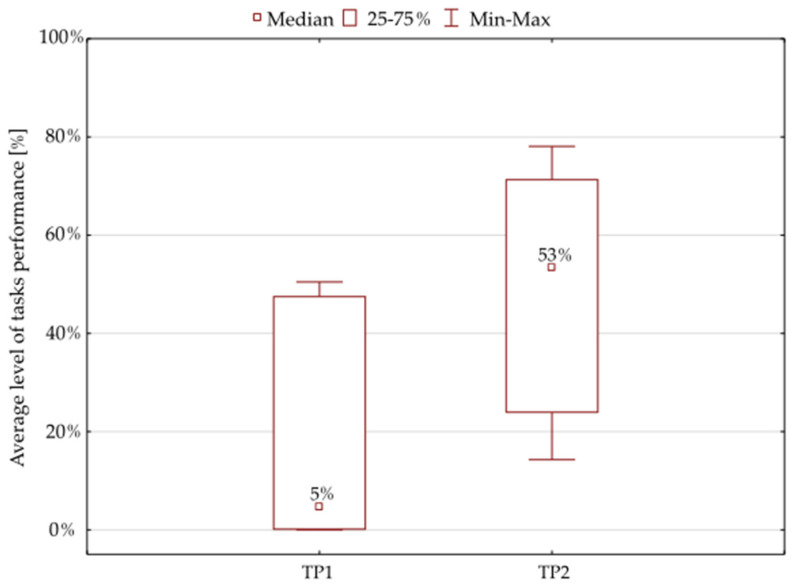
Average percentage of the tasks performed by the whole study group. Note: TP1—percentage of tasks performed during the initial tests, TP2—percentage of tasks completed during the final tests.

**Table 1 ijerph-19-03081-t001:** Study group characteristics.

TBI					ABI
Participant	P1	P2	P3	P4	P5
Diagnosis	CT	CT	CT	CT	HRSS
Age [years]	25	31	26	40	67
Time [years]	5	3.5	8	5	1
CRS-R	16	18	22	9	22
Conscious state	MCS	eMCS	eMCS	UWS	eMCS

Note: TBI—traumatic brain injury, ABI—acquired brain injury, CT—cerebrocranial trauma, HRSS—haemorrhagic right-sided stroke; Time—interval of time from the critical event to the beginning of the study, P—patient, MCS—minimally conscious state, UWS—unresponsive wakefulness syndrome, eMCS—emergence from the minimally conscious state.

**Table 2 ijerph-19-03081-t002:** Comparisons of durations of the patients’ active work, before and after the oculomotor training course.

	P1	P2	P3	P4	P5	Me	Q_1_	Q_3_	*p*
B1	10	40	35	30	30	30	30.0	35.0	0.043115 *
B2	30	60	45	45	50	45	45.0	50.0	

Note: B1—active work duration (minutes)—initial tests, B2—active work duration (minutes)—final tests, ME—median value, Q_1_—lower quartile, Q_3_—upper quartile, * *p* < 0.05000.

**Table 3 ijerph-19-03081-t003:** Wilcoxon matched pairs test for the rates of performed tasks before and after training.

	N	Me	Q_1_	Q_3_	*p*
TP 1	5	0.047619	0.00000	0.476190	0.043115 *
TP 2	5	0.533333	0.238095	0.714286	

TP1—the number of tasks performed by the whole study group before training, TP2—the number of tasks performed by the whole study group after training, N—the number of participants, Me—median values, Q_1_—lower quartile, Q_3_—upper quartile, * *p* < 0.05000.

## Data Availability

The data presented in this study are available on request from the corresponding author.

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
