# Peer review of "Implementing New Technologies to Improve Visual–Spatial Functions in Patients with Impaired Consciousness"

_ijerph, 2022, doi:10.3390/ijerph19053081_

Round 1
Reviewer 1 Report
The paper by Kujawa and colleagues reports the research findings regarding participants receiving an oculomotor training course aimed at the therapy of visual-spatial functions. Over six weeks, the subjects solved different visuo-spatial tasks with the use of an eye tracker. The findings present evidence of the oculomotor training effectiveness based on a longer duration of the work with the eye tracker and improved visual-spatial functions. I think that the study is interesting and well conducted and proposes a novel way to improve patients' cognitive abilities and quality of life. However, I do have a few suggestions that I hope may improve the quality of the manuscript.
Major points:
- In my opinion, more information on patients and their characteristics are necessary to better delineate who can benefit more from the training. For instance, I suggest adding some information on the time elapsed since the accident, the lesion site, its extension, etc.
- I think that the paper is missing a section dedicated to the eye-tracking training. I see that the authors describe the apparatus (i.e., the C-Eye Pro eye tracker), but some information on how the training has been conducted is required to favor future replication attempts. For instance, the authors say the training lasted 6 weeks, but how many sessions per week were conducted? How long did each session last? Who conducted the training (e.g., medical doctor, therapist, etc). Please add all the relevant specifics.
- In line 259, the authors claim: "Moreover, better focussing on selected objects was observed among the patients after the training of ocular muscles. In addition, improved smooth pursuit was demonstrated in each subject after the research. Reduced numbers of saccadic pursuit movements were also observed, which suggests improved eye movement control in the patients following the oculomotor training course." Are there any analyses supporting these findings?
- In my opinion, a final section where the authors acknowledge the limitations of the present study is necessary. In particular, the authors should acknowledge the lack of control conditions (either a control group or a control intervention), the very small sample size included in their study, and all the relevant limitations that may hamper the generalizability of their findings.
Minor points:
- Line 88: typo "0{>ETs"
- Maybe it is worth considering adding a figure to represent a trial example of the different visuo-spatial tasks.
Author Response
We would like to sincerely thank the Reviewer for the positive feedback and helpful comments. We strongly focused our efforts on the points made in your letter. We would like to respond to your opinion based on our careful revision, point by point below.
Reviewer: In my opinion, more information on patients and their characteristics are necessary to better delineate who can benefit more from the training. For instance, I suggest adding some information on the time elapsed since the accident, the lesion site, its extension, etc.
Answer: thank you for that suggestion. We added the data in the Table 1. with the time elapsed since the accident. Information about the etiology of the brain injury is presented in Table 1. In addition, we added information in the Characteristics of the Study Group section that explains how patients were selected for the group in which the exact location or extent of the injury was secondary to the ability to work with the eye tracker. The following information has been added: Inclusion criteria were: consent of the patient’s guardian to participate in the study, person after completion of standard medical care, lack of verbal and motor communication with the environment, brain damage of different ethology, at least one functioning eye-ball.
Reviewer: I think that the paper is missing a section dedicated to the eye-tracking training. I see that the authors describe the apparatus (i.e., the C-Eye Pro eye tracker), but some information on how the training has been conducted is required to favor future replication attempts. For instance, the authors say the training lasted 6 weeks, but how many sessions per week were conducted? How long did each session last? Who conducted the training (e.g., medical doctor, therapist, etc). Please add all the relevant specifics.
Answer: thank you for that comment. According to your suggestion we have added the new section, called Oculomotor training (Section 2.4).
Reviewer: In line 259, the authors claim: "Moreover, better focusing on selected objects was observed among the patients after the training of ocular muscles. In addition, improved smooth pursuit was demonstrated in each subject after the research. Reduced numbers of saccadic pursuit movements were also observed, which suggests improved eye movement control in the patients following the oculomotor training course." Are there any analyses supporting these findings?
Answer: thank you for that comment. The phrase in the manuscript referred to by the reviewer was the result of observations made while working with the patient. However, the suggestion in the final part of this sentence (...which suggests improved eye movement control in the patients following the oculomotor training course...) is also supported by the results of studies by other authors showing the importance of oculomotor training using eye tracking technology. This is indicated for example by Murray et al. (2021), presenting the improvement of saccade parameters after oculomotor training applied.
Murray, N.; Hunfalvay, M.; Roberts, C.M.; Tyagi, A.; Whittaker, J.; Noel, C. Oculomotor Training for Poor Saccades Improves Functional Vision Scores and Neurobehavioral Symptoms. Archives of rehabilitation research and clinical translation 2021, 3, doi:10.1016/j.arrct.2021.100126.
This information has been added in the Discussion section (see lines 333-334).
Reviewer: In my opinion, a final section where the authors acknowledge the limitations of the present study is necessary. In particular, the authors should acknowledge the lack of control conditions (either a control group or a control intervention), the very small sample size included in their study, and all the relevant limitations that may hamper the generalizability of their findings.
Answer: thank you for that comment. We have added the Limitation section; in this section we have shown the limitations related to our study.
Minor points:
Reviewer: Line 88: typo "0{>ETs"
Answer: thank you for pointing out the error. The characters before the abbreviation ET should not appear here
Reviewer: Maybe it is worth considering adding a figure to represent a trial example of the different visuo-spatial tasks.
Answer: thank you for that comment. In order to better visualize the trials performed by the study participants, we have added figures showing examples of tasks from each category. This material has been added as the Supplementary file (Figure 1S: categories of visual-spatial function testing).
In addition, in order to be accurate in the publication of statistical data, the values of the lower and upper quartiles were introduced in Tables 2 and 3 instead of the reported sd values
Reviewer 2 Report
In this study, the authors present data that show the effectiveness of oculomotor training in improving the visual-spatial functioning of patients that have impaired consciousness.
Improving functioning of patients with impaired consciousness is an important problem that needs addressing. However, there are several details that the paper is missing which I think the readers would be interested in knowing, and will also help in improving clarity of the manuscript.
Major points
- The authors mention 'oculomotor training'. However, there is no description of what the training involved. That seems like an important missing detail.
- The authors use two main metrics for quantifying performance before and after oculomotor training: duration of work and percentage of successful tasks performed. Again, there are several details missing.
- How many trials were performed for each category (A-G)? Presently the authors only present % of trials successfully completed (I think? the labels in Figure 2 could be better).
- The duration comparisons presented in table 2 are barely significant, and such is not a very strong measure of improvement. I am not convinced that this indicates improvement (see point 3 below).
- From what I understand, the patient was given an instruction, and had to select the correct answer by moving his/her gaze to the correct choice on the lower part of the screen. How long do patients take to complete a particular task before and after oculomotor training? I think that is an important metric that I would be interested in knowing. For example, does gaze meander before reaching the target, or does it take a near direct path to the target? Does this improve after oculomotor training? If gaze meanders, maybe a metric that measures distance traveled before reaching target.
- One thing I am not clear about. Do these patients have differing oculomotor performance before training? I think that a performance benchmark needs to be established. That way improvement/changes post training can be normalized to the pre-training oculomotor performance. If oculomotor performance of the patients wildly differs before the training, then the data might be hard to interpret post-training. If the authors are already doing this, they should make it clear in the manuscript.
- The p-value presented in table 2 and 3 is the same. This seems like a typo. Please correct that. The value in table 3 is clearly incorrect.
- The Discussion section is too tedious. It could be shortened significantly. For example, details of how the retina works (lines 287-299) is probably not necessary or can be significantly shortened. The whole discussion section could be reduced to half of what it is now.
Minor points
Labeling of figure axis needs to be improved. For example, Figure 2 y-axis could be '% of tasks completed'.
Author Response
We would like to sincerely thank the Reviewer for the positive feedback and helpful comments. We strongly focused our efforts on the points made in your letter. We would like to respond to your opinion based on our careful revision, point by point below.
Major points
Reviewer: The authors mention 'oculomotor training'. However, there is no description of what the training involved. That seems like an important missing detail.
Answer: thank you for that comment. We have added the new section (2.4 Oculomotor training), describing what the oculomotor training used in the project involved.
Reviewer: The authors use two main metrics for quantifying performance before and after oculomotor training: duration of work and percentage of successful tasks performed. Again, there are several details missing.
- How many trials were performed for each category (A-G)? Presently the authors only present % of trials successfully completed (I think? the labels in Figure 2 could be better).
- The duration comparisons presented in table 2 are barely significant, and such is not a very strong measure of improvement. I am not convinced that this indicates improvement (see point 3 below).
Answer: thank you for that comment. In each training session, tasks from each category were performed. This means that during the six-week training period, each task category was performed a total of 6 times. Information about the number of tasks has been added in the Visual-Spatial Function Testing section (2.3). In this study we focused on comparing the results obtained before and after the six-week oculomotor training (initial tests vs final tests).
The results in Table 2 show the patient's time actively working with the eye tracker before and after the six weeks of oculomotor training. Statistical analysis shows that the observed change is statistically significant. To emphasize this fact, we added an asterisk next to the p value=0.043115. Similar information has been added in Table 3.
Reviewer: From what I understand, the patient was given an instruction, and had to select the correct answer by moving his/her gaze to the correct choice on the lower part of the screen. How long do patients take to complete a particular task before and after oculomotor training? I think that is an important metric that I would be interested in knowing. For example, does gaze meander before reaching the target, or does it take a near direct path to the target? Does this improve after oculomotor training? If gaze meanders, maybe a metric that measures distance traveled before reaching target.
Answer: thank you for that comment. We highly appreciate the reviewer's question about the time it takes for the patient to search for the answer before he / she indicates the correct answer with their eyes. We did not intend to measure such time in our study, but recognizing the potential importance of this parameter as well as the importance of distance travelled before reaching target we decided to include this information in the Conclusion section as an implication for the future studies. Answering the question of how long do patients take to complete a particular task before and after oculomotor training will help clarify which category of tasks improve the most and which require the most effort from the patient.
Reviewer: One thing I am not clear about. Do these patients have differing oculomotor performance before training? I think that a performance benchmark needs to be established. That way improvement/changes post training can be normalized to the pre-training oculomotor performance. If oculomotor performance of the patients wildly differs before the training, then the data might be hard to interpret post-training. If the authors are already doing this, they should make it clear in the manuscript.
Answer: thank you for that comment. Indeed, we are aware that the oculomotor performance of the patients (P1 - P5) before the start of the six-week oculomotor training varied. It is, of course, a result of the severity of brain damage, the consequences of which tend to manifest in individually variable ways. Therefore, we see our data in an individual context but also, in order to make some generalizations, we decided to use simple statistical methods, comparing the change in total scores before training versus total scores after training (initial test vs final test). This information is shown in the Statistical analysis section.
Reviewer: The p-value presented in table 2 and 3 is the same. This seems like a typo. Please correct that. The value in table 3 is clearly incorrect.
Answer: thank you for that comment. A Wilcoxon rank-sum test was performed for both variables, comparing the five-element samples with the repeated survey. It is important to note that therefore the original measurements are not analyzed but only the ranks assigned to them, therefore for such a small sample two different variables showed the same rank order. This causes the p value to be the same in both tables.
In addition, in order to be accurate in the publication of statistical data, the values of the lower and upper quartiles were introduced in Tables 2 and 3 instead of the reported sd values
Reviewer: The Discussion section is too tedious. It could be shortened significantly. For example, details of how the retina works (lines 287-299) is probably not necessary or can be significantly shortened. The whole discussion section could be reduced to half of what it is now.
Answer: thank you for that comment. We have carefully reviewed the Discussion section and removed some parts of it. In some places, the results of our own research require, in our opinion, a more extensive presentation of the neurological mechanisms, so we considered them worth leaving in the manuscript and will make it easier for the reader to follow and understand the text We hope this will be more acceptable and more interesting for the reader
Minor points
Reviewer: Labeling of figure axis needs to be improved. For example, Figure 2 y-axis could be '% of tasks completed'.
Answer: thank you for that comment. At the suggestion of a reviewer, we have made some corrections; we have corrected the y-axis description and placed a legend to the right of the graph to make the entire graphic more readable. We hope that this will improve the clarity of the graph.
Round 2
Reviewer 2 Report
Authors have answered my questions. I have no further comments.